# Mitigating Effects of *Tanacetum balsamita* L. on Metabolic Dysfunction-Associated Fatty Liver Disease (MAFLD)

**DOI:** 10.3390/plants13152086

**Published:** 2024-07-27

**Authors:** Rositsa Mihaylova, Reneta Gevrenova, Alexandra Petrova, Yonko Savov, Dimitrina Zheleva-Dimitrova, Vessela Balabanova, Georgi Momekov, Rumyana Simeonova

**Affiliations:** 1Department of Pharmacology, Pharmacotherapy and Toxicology, Faculty of Pharmacy, Medical University of Sofia, Sofia 1000, Bulgaria; rmihaylova@pharmfac.mu-sofia.bg (R.M.); aleksandrapetrova91@yahoo.com (A.P.); gmomekov@pharmfac.mu-sofia.bg (G.M.); rsimeonova@pharmfac.mu-sofia.bg (R.S.); 2Department of Pharmacognosy, Faculty of Pharmacy, Medical University of Sofia, Sofia 1000, Bulgaria; dzheleva@pharmfac.mu-sofia.bg (D.Z.-D.); vbalabanova@pharmfac.mu-sofia.bg (V.B.); 3Institute of Emergency Medicine “N. I. Pirogov”, Bul. Totleben 21, Sofia 1000, Bulgaria; yonko_savov@hotmail.com

**Keywords:** metabolic syndrome, obesity, metabolic-associated fatty liver disease, oxidative stress, antioxidants, polyphenols, *Tanacetum balsamita*

## Abstract

The metabolic syndrome and its associated co-morbidities have been recognized as predisposing risk factors for the development of metabolic-associated fatty liver disease (MAFLD). The present study reports on the beneficial effect of the *Tanacetum balsamita* methanol-aqueous extract (ETB) at 150 and 300 mg/kg bw on biochemical parameters related to oxidative stress, metabolic syndrome, and liver function in rat animal models with induced MAFLD. ETB was found to be non-toxic with LD50 > 3000 mg/kg and did not affect cell viability of hepatic HEP-G2 cells in a concentration up to 800 μg/mL. The pathology was established by a high-calorie diet and streptozotocin. Acarbose and atorvastatin were used as positive controls. At the higher dose, ETB reduced significantly (*p* < 0.05) the blood glucose levels by about 20%, decreased lipase activity by 52%, total cholesterol and triglycerides by 50% and 57%, respectively, and restored the amylase activity and leukocytes compared to the MAFLD group. ETB ameliorated oxidative stress biomarkers reduced glutathione and malondialdehyde in a dose-dependent manner. At 300 mg/kg, the beneficial effect of the extract on antioxidant enzymes was evidenced by the elevated catalase, glutathione peroxidase, and superoxide dismutase activity by 70%, 29%, and 44%, accordingly, compared to the MAFLD rats. ETB prevents the histopathological changes related to MAFLD. ETB, rich in 3,5-dicafeoylquinic, chlorogenic, and rosmarinic acids together with the isorhamnetin- and luteolin-glucoside provides a prominent amelioration of MAFLD.

## 1. Introduction

Metabolic syndrome (MS) comprises a cluster of metabolic defects including hypertension, insulin resistance, visceral obesity, fatty liver, and atherogenic dyslipidemia [1]. A clinically related pathology is the metabolic-associated fatty liver disease (MAFLD), a novel terminology introduced in 2020 to provide a more accurate description of fatty liver disease linked to metabolic dysfunction, replacing the outdated term nonalcoholic fatty liver disease (NAFLD) [2]. The co-occurrence of metabolic risk factors such as overweight or obesity, type 2 diabetes, and insulin resistance, alongside the associated deviations in the biochemical parameters (high triglyceride and/or LDL cholesterol level and high blood glucose levels) differentiates MAFLD from NAFLD and accounts for the growing prevalence of MAFLD among the non-communicable diseases worldwide. The close clinical association between MAFLD and metabolic syndrome is determined by their common pathophysiological background, in which oxidative stress plays a central role. Perturbations in the cellular redox state have been recognized as both a triggering and exacerbating factor in the development of various metabolically related chronic diseases including obesity, type 2 diabetes (DT2), dyslipidemia, hypertension, hepatic dysfunction, etc. [3,4]. As there is no suitable, safe, and effective drug yet available in the clinic that covers the entire spectrum of MAFLD and its underlying conditions, the pursuit of new compounds for the effective primary prevention or treatment of MAFLD continues [5]. As a source of potent natural antioxidants, phytotherapy holds great promise for the effective restoration of cellular redox homeostasis and has been extensively explored as an alternative or adjuvant means to conventional therapy. Polyherbal mixtures, plant extracts, and/or biologically active phytoconstituents derived from them can be combined with conventional drugs to achieve stronger synergistic therapeutic effects or to reduce the intensity or manifestations of drug-induced adverse effects [3]. Within the context of metabolic disorders, numerous plant extracts with high phenolic contents and ROS neutralizing capacity have been evaluated both in vitro and in vivo for their protective properties against oxidative morphological and functional organ damage [6,7].

The genus *Tanacetum* has been traditionally used to manage several oxidative stress-related diseases such as diabetes, hypercholesterolemia, and nerve system-related conditions. Various in vitro and in vivo studies have corroborated the ethnopharmacological uses of *Tanacetum* spp. as a traditional antioxidant remedy [8]. Crude extracts of *Tanacetum* taxa and isolated metabolites have demonstrated antidiabetic and antioxidant activities [8]. However, only four taxa have been evaluated for their in vitro antidiabetic activity, namely, *T. praeteritum*, *T. haussknechtii*, *T. balsamita*, and *T. nubigenum*. So far, *T. nubigenum* is the only species assessed for its in vivo antidiabetic activity [8].

The less studied species *T. balsamita* (costmary) is a well-known spice used as a flavor, carminative, and cardiotonic in the Mediterranean, Balkan, and South American countries [9]. The plant is distributed in the southeast of Europe and southwest of Asia and widely naturalized throughout the whole world [10,11]. Major chemical constituents identified in the genus *Tanacetum* L. are acylquinic acids, sesquiterpenes, sesquiterpene lactones, and methoxylated flavonoids [12]. Previously, more than 100 secondary metabolites including methoxylated flavonols and flavones, acylquinic acids analogues, hydroxybenzoic and hydroxycinnamic acids derivatives, and their glycosides were annotated/dereplicated in the costmary leaves, flower heads, and roots. The acylquinic acids chlorogenic, 3,5-diCQA, and 4,5-diCQA acid together with luteolin- and jaceosidin-hexuronide and aglycones nepetin and jaceosidin dominated the leaf and root profilings and are most probably distinctly accountable for the reported antioxidant activity of the *Tanacetum* species [12,13,14,15]. Costmary leaf extract actively scavenged the free radicals DPPH and ABTS, and showed a high reducing power in FRAP and CUPRAC assays and strong chelating ability [12]. Given their multimodal antioxidant, anti-inflammatory, and antidiabetic properties [13], we argue that such polyphenol-rich species would exert a potent modulating effect on the pathological processes accompanying the metabolic syndrome. In view of this, the present study is dedicated to the in vivo evaluation of the antidiabetic, hepatoprotective, and antioxidant properties of *Tanacetum balsamita* L. (Asteraceae) in rat models with induced MAFLD. The potential mitigating effects of the *Tanacetum balsamita* leaf extract (ETB) on the in vivo antioxidant status and lipid/glucose profiles were assessed in different dosing regimens and against positive control groups treated with the reference drugs acarbose and atorvastatin.

## 2. Results and Discussions

### 2.1. UHPLC-DAD Analysis

An UHPLC-DAD method for the quantitative determination of the main compounds in the extract from *T. balsamita* (ETB) was developed. A total of 16 analytes, including 7 acylquinic acids (1, 2, 8, 9, 12, 13, and 14), rosmarinic acid (11), and 8 flavonoids (3, 4, 5, 6, 7, 10, 15, and 16) were determined in ETB. An UHPLC-DAD chromatogram of ETB is presented in Figure 1. Analytical characteristics of the UHPLC-DAD method are presented in Table 1. The linear regression for the calibration curves showed good linear relationships with r^2^ > 0.9890 within the test ranges. The content of the assayed compounds is revealed in Table 2.

The main compound in ETB was 3,5-diCQA (9), followed by chlorogenic acid (2), rosmarinic acid (11), and 3,4-diCQA (8). In addition, based on UV-spectra, compounds 12, 13, and 14 were identified as acylquinic acids (AQAs) and quantified. Isorhamnetin 3-*O*-glucoside was the dominant flavonoid in ETB, followed by orientin (3), luteolin (15), hyperoside (5), and luteolin 7-*O*-glucoside (6) (Table 2). The content of the acylquinic acids reached up to 12.13% in the lyophilized ETB, while the amount of the analyzed flavonoids was 2.32%.

The presence of the above-mentioned compounds in costmary was reported previously, but without thorough quantitative data [12,13]. However, a former HPLC-DAD analysis identified four major flavonoids in *T. balsamita* extract as quercetin, apigenin 7-*O*-glucoside (cosmosiin), luteolin 7-*O*-glucoside, and luteolin 3-methyl ether (chrysoeriol) [16]. Additionally, cichoric acid was determined in that study as the prevailing compound in the costmary, being present at 3.33 g/100 g extract. In contrast, cichoric acid was not found in our study. The values of chlorogenic and rosmarinic acid determined by Bazek and co-workers were considerably lower than those found in our study, while a lesser amount of luteolin 7-*O* glucoside was quantified.

### 2.2. In Vitro Cytotoxicity Study

The in vitro effect of the Tanacetum balsamita extract (ETB) on cell proliferation was assessed against hepatic human-derived HEP-G2 cells following a 72 h exposure. As indicated by Figure 2, treatment with the ETB had no influence on cell growth and viability even at the highest concentration tested (800 μg/mL). Under microscopic examination, no morphological changes in cell appearance regarding size, form, membrane integrity, swelling, detachment, etc., were detected.

### 2.3. In Vivo Acute Toxicity Study

An acute toxicity study was conducted to evaluate the single-dose toxicity of the *T. balsamita* extract (ETB) and estimate the LD50 range. Administered at a dose of 3000 mg/kg, the tested extract did not cause toxic effects and death in the experimental animals. No changes in behavior or food and water intake were observed. Therefore, a dose of 3000 mg/kg b.w. was non-toxic to the tested extract administered once orally to male Wistar rats, and it could be determined that LD50 > 3000 mg/kg.

Animals that survived the acute toxicity were euthanized on day 15 of the acute administration of the ETB, after anesthesia with ketamine/xylazine (80/10 mg/kg, i.p.). Internal organs were inspected (organ color, consistency, neoplasms, etc.) and no macroscopic changes in the organs were noted.

### 2.4. In Vivo Chronic Study

The potential modulating effects of the *Tanacetum balsamita* extract on the antioxidant status and lipid/glucose profile were evaluated in male healthy rats and rats with induced metabolic dysfunction-associated fatty liver disease (MAFLD) in different dosing regimens and in a comparative manner to the reference drugs acarbose and atorvastatin. For the following experiments, doses of 1/20 and 1/10 of the LD50 were chosen, i.e., 150 mg/kg and 300 mg/kg with which the experimental animals were treated orally for 8 weeks.

#### 2.4.1. Body Weight Changes

The change in body weight of the experimental animals is reflected in Figure 3a. At the beginning of the experiment, all the animals included in the study groups had approximately similar body weights between 155 and 165 g. In all groups with induced metabolic disorders, the change in body weight was insignificant. Even in the control MAFLD group, at the end of the experimental period, a slight decrease in body weight was observed compared to the beginning of the experiment (Figure 3a).

Most subjects with metabolic syndrome are obese, but it should be noted that a significant proportion of MAFLD patients are not obese [5,17,18,19,20,21,22,23,24], a fact that was also confirmed in our experiment. The most pronounced increase in body weight by 13% compared to the beginning of the experiment was observed in the group treated only with the high dose of ETB (ETBhd).

After the administration of STZ on day 21, a sharp increase in blood sugar was observed in all groups fed the high-calorie diet, most prominently in the MAFLD and MLD + Ats groups, where the increase in blood sugar compared to the beginning of the experiment was about 65% (Figure 3b). In the MAFLD groups treated with acarbose, ETBld, and ETBhd, the increase in blood sugar was 39%, 48%, and 40%, respectively, compared to the beginning of the experiment. In the automatic measurement of glucose in the serum of the animals (Table 3) after euthanasia at the end of the experiment, values close to those measured with the test strips were found. Table 3 shows that acarbose reduced the blood sugar level to the greatest extent by 25% compared to the MAFLD group, and ETB administered in both doses reduced the blood sugar level statistically significantly by about 20% compared to the MAFLD group. The hypoglycemic effects of acarbose and the Tanacetum extract were comparable, probably due to their identical mechanism of action. Acarbose is a hypoglycemic drug that decreases blood glucose levels by inhibiting the enzyme α-glucosidase. Similarly, the antihyperglycemic effects of the genus *Tanacetum* have been attributed to its α-glucosidase inhibitory activity in a number of studies [8,15,25]. Gevrenova et al. (2023) reported that root methanolic-aqueous extract of *T. balsamita* had good α-glucosidase and α-amylase inhibitory effects with values of 0.71 ± 0.07 mmol acarbose/g and 0.43 ± 0.02 mmol acarbose/g, respectively [12].

The antidiabetic effects of *Tanacetum* spp. could be attributed to a variety of biologically active substances (BAS), especially sesquiterpene lactones, acylquinic acids, phenolic acids, hexosides, flavones-glucuroniodes, and methoxylated flavones and flavonols. Chlorogenic acid, or 5-caffeoylquinic acid, has been identified in some *Tanacetum* extracts and was also found as one of the main polyphenolic components of the herein studied *T*. *balsamita* extract. Clinical data reported the ability of this compound to decrease fasting blood glucose significantly when consumed for 12 weeks. Chlorogenic acid can improve glucose homeostasis by up-regulating the expression and translocation of glucose transporter type 4 (GLUT-4) in the skeletal muscle of mice models [14]. Clifford et al. stated that acylquinic acids (chlorogenic acids) can reduce the risk of developing type 2 diabetes and cardiovascular diseases [26].

The triglyceride levels (Figure 3c) increased manifold in all groups with induced MAFLD, especially after the fourth week, and to the highest extent in the control MAFLD group, where at the end of the experiment, the measured triglyceride levels were 567% higher than the baseline level. In the groups treated with acarbose, ETBld, and ETBhd, the triglyceride concentration at the end of the experiment was, respectively, 275%, 280%, and 140% higher compared to the starting point.

#### 2.4.2. Serum Biochemical and Hematological Parameters

The lowest peak in triglyceride levels was seen in the MAFLD group treated with atorvastatin, where a nearly 2-fold increase of 109% was observed compared to the beginning of the experiment.

Although the primary therapeutic effect of statins is on LDL cholesterol and not triglyceride levels, in our experiment, atorvastatin produced the greatest 65% reduction in triglyceride levels compared to the MAFLD group (Table 3), followed by ETBld (43%) and ETBhd (57%). Acarbose exerted the weakest effect on triglyceride levels, although a statistically significant reduction by 36% was observed compared to the MFLD group.

Lipase is an enzyme that promotes the degradation of triglycerides. It could be speculated that because of the higher level of triglycerides in MAFLD rats in the present study, the activity of this enzyme was 190% higher than in the control rats. Consistent with their lowering effect on triglyceride levels, atorvastatin and the high-dose ETB also reduced lipase activity by 62% and 52%, respectively, compared to the MAFLD group. Acarbose and the low dose of ETB exhibited a weaker effect on this enzyme activity by reducing it by ca. 30% compared to the MAFLD group. Despite the observed decrease in lipase activity, its values in all MAFLD groups remained higher than the upper reference limit, which is 14 U/L.

The dynamics of changes in total cholesterol levels were similar to the fluctuations observed in triglyceride levels (Figure 3d). Compared to the beginning of the experiment, the highest elevation in cholesterol levels at the end of the period was registered in the MAFLD group (by 233%), followed by the cholesterol level in the metabolic animals treated with acarbose (by 167%). The increase in cholesterol level at the end of the experimental period was prevented to the greatest extent by atorvastatin and ETBhd by approximately 50%.

According to the data in Table 3, the HC-HF diet and diabetes led to a statistically significant increase in cholesterol levels in the MAFLD group by 202% compared to the control. The repeated administration of atorvastatin, and both the ETBld and ETBhd to animals with induced metabolic syndrome, was able to induce a statistically significant decrease in serum cholesterol levels by about 45% compared to the MAFLD group.

Statins are the first-line drugs for lowering cholesterol levels, especially low-density lipoprotein-cholesterol (LDL-C) levels. Their primary mechanism of action involves inhibition of the expression and activity of the hepatic 3-hydroxy-3-methylglutaryl coenzyme A reductase (HMGR) enzyme and a subsequent upregulation in the density and function of the LDL receptor (LDLR) on the hepatocyte membrane [27]. In addition, they can exert a secondary modulating effect on VLDL and triglyceride levels resulting from their primary mechanism of action [28,29].

Herein, the quantitative analysis of *T. balsamita* phytochemicals revealed the preponderance of acylquinic acids, including 3, 5-dicafeoylquinic and chlorogenic acid, along with rosmarinic acid, isorhamnetin 3-*O*-glucoside, and luteolin 7-*O*-glucoside. Antidyslipidemic effects have been established for many of these compounds. Recently, dietary polyphenols have received much attention in some pathology prevention due to their protective effects [30,31].

An in vivo study conducted by Sun et al. revealed that the presence of quercetin and isoquercitrin in onion reduced plasma LDL-C levels by inhibiting the HMGR enzyme in cholesterol biosynthesis and inducing the expression of lipoprotein receptor (LDLR) on hepatocytes [27]. The authors suggested that quercetin and its glycoside lowered LDL-C levels by several other mechanisms, including inhibition of cholesterol resorption via the NPC1L1 transporter, a critical protein in dietary cholesterol uptake, and attenuating proprotein convertase subtilisin/kexin type 9 (PCSK9) secretion which is of key importance for the recycling of LDLR [27].

In our study, the amylase activity was lower by 32% in the MAFLD-induced rats compared with the control group. Treatment with atorvastatin, acarbose, and low- and high-dose ETB lead to a restoration of the amylase enzyme activity. Low serum amylase is associated with an increased risk of metabolic abnormalities. Hypoamylasemia has been reported in certain common metabolic conditions such as obesity, diabetes (regardless of type), and metabolic syndrome, all of which appear to have a common etiology of insufficient insulin action due to insulin resistance and/or diminished insulin secretion [32].

The metabolic disturbances induced by the high-calorie diet and administration of streptozotocin and fructose led to a deterioration of liver function. The induced organ damage manifested in an increased level of total bilirubin and transaminase activity by around 100%, and a decrease in albumin by 28%, compared to the control values (Table 3). Neither the administered drugs atorvastatin and acarbose nor the two doses of the extract were able to normalize the activity of transaminases, and they remained high above the reference values. The observed abnormalities in the liver biochemistry are a sign of serious disturbance in both the liver’s synthetic function and its detoxification capabilities.

**Table 3 plants-13-02086-t003:** Serum biochemical parameters in all treated groups.

Blood Biochemistry	Controls	ETBld	ETBhd	MAFLD	MLD + Ats	MLD + Acarb	MLD + ETBld	MLD + ETBhd	Ref. Range (Rats)
GLU mmol/L	6.95 ± 0.82	7.02 ± 0.77	7.25 ± 0.76	10.4 ± 0.4 *^#^	8.96 ± 0.9 ^#^	7.8 ± 0.79	8.3 ± 0.68	8.2 ± 0.56	4.46–7.24 [33]
TP g/L	58.1 ± 2.2	40.7 ± 3.1	59.6 ± 2.6	59.1 ± 3.6	58.5 ± 3.8	61.3 ± 5.6	60.2 ± 6.3	60.8 ± 5.7	51.1–64.6 [33]
ALB g/L	35.8 ± 1.8 ^#^	35.2 ± 1.7 ^#^	33.4 ± 2.2 ^#^	25.8 ± 3.1 *	27.6 ± 2.2	26.6 ± 2.8 ^#^	26.2 ± 3.2 ^#^	24.8 ± 2.6 ^#^	26.9–34.6 [33]
T-BIL µmol/L	4.9 ± 0.3	4.6 ± 0.4	5.2 ± 0.2	9.6 ± 0.6 *	8.9 ± 0.2	8.2 ± 0.2	7.2 ± 0.3 +	7.4 ± 0.2 ^+^	1.7–8.5 [34]
D-BIL µmol/L	2.7 ± 0.03	1.8 ± 0.04	1.7 ± 0.02	2.4 ± 0.01 *	2.8 ± 0.03	2.7 ± 0.04	3.1 ± 0.02	2.8 ± 0.02	0–5.13 [34]
ASAT U/L	113 ± 4.5	123.7 ± 5.2 *	122.3 ± 3.6 *^+^	221 ± 4.1 *	180.5 ± 6.8 ^+#^	168.8 ± 7.2 ^+#^	165.7 ± 6.3 ^+#^	169 ± 5.8 ^+#^	60–139 [33]
ALAT U/L	43.8 ± 2.2	40.5 ± 3.1	46.8 ± 3.3	89.2 ± 3.4 *	107 ± 4.5 ^+#^	87 ± 6.3 ^#^	83.1 ± 7.2 ^#^	94.8 ± 7.7 ^#^	19–47 [33]
AMYL U/L	1390 ± 112	1663 ± 208	1495 ± 109	951.5 ± 93 *	1336 ± 102 ^+^	1287 ± 98.8 ^+^	1501 ± 52.1 ^+^	2012 ± 126 ^+^	1416–3161 [35]
CHOL mmol/L	1.39 ± 0.02	1.58 ± 0.01	1.62 ± 0.03	4.2 ± 0.02 *	2.16 ± 0.02 ^+^	3.31 ± 0.04 ^+^	2.4 ± 0.03 ^+^	2.2 ± 0.02 ^+^	0.68–1.77 [33]
TRIG mmol/L	0.42 ± 0.01 ^#^	0.43 ± 0.02 ^#^	0.45 ± 0.02 ^#^	2.8 ± 0.03 *	0.98 ± 0.04 ^+^	1.8± 0.02 +	1.6 ± 0.03 ^+^	1.2 ± 0.03 ^+^	0.23–0.99 [33]
UA µmol/L	83.6 ± 4.4	72.3 ± 2.3 *	77.2 ± 3.7	120 ± 6.7 *	88.3 ± 3.3	49.3 ± 7.8 *	55.1 ± 8.2 *	68.2 ± 4.4	0.1–760 [33]
Lipase U/L	13.2 ± 0.42	12.6 ± 0.51	11.3 ± 0.31	38.4 ± 0.22 *^#^	14.7 ± 0.31 ^+^	27.4 ± 0.18 ^+#^	26.4 ± 0.14 ^+#^	18.6 ± 0.21 ^+#^	7–14 [36]

* *p* ≤ 0.01 vs. control; ^+^
*p* ≤ 0.01 vs. MAFLD; ^#^
*p* ≤ 0.01 vs. ref. range. Results are expressed as mean ± SD (n = 5). The significance of the data was assessed using the Bonferroni-corrected Mann–Whitney U test. Values of *p* ≤ 0.01 were considered statistically significant. Abbreviations: GLU, glucose; TP, total protein; ALB, albumin; ASAT, aspartate aminotransferase; ALAT, alanine aminotransferase; AMYL, amylase; UA, uric acid; T-BIL, total bilirubin; D-BIL, direct bilirubin; CHOL; cholesterol; TRIG, triglicerides.

Changes in hematological parameters are presented in Table 4. It is noteworthy that all parameters were within the acceptable reference limits for rats, except for leukocyte and platelet count. A higher white blood cell count was found in all groups with induced metabolic disorders, which is expected and related to the general inflammation in this condition. Only in the group treated with the high dose of ETB, a 34% lower number of leukocytes was observed compared to the control MAFLD group.

In all experimental groups, without exception, the platelet level was statistically significantly lower than the reference limits for rats, which is probably a characteristic of the animals from our breeding center.

#### 2.4.3. Oxidative Stress Biomarkers

The role of oxidative stress has been proven in the pathology of dyslipidemia and diabetes mellitus and is often accompanied by a variety of other metabolic disturbances such as elevated levels of uric acid, hypertension, and endothelial dysfunction [37].

Oxidative stress is also elevated in the obesity state and promotes obesity-induced metabolic dysfunction by increasing reactive oxygen species (ROS) in adipose tissues. It is defined as an imbalance between ROS production and enzymatic or non-enzymatic antioxidants’ capacity. Excessive ROS production can further stimulate the oxidation of LDL-C contributing to atherosclerotic plaque formation. The free radicals and non-radical species can damage lipid, protein, and DNA components activating various pathological pathways in the development of a myriad of diseases and their associated complications [4].

In the present study, MAFLD is characterized by a marked 36% increase in the production of malondialdehyde (MDA), a byproduct of polyunsaturated fatty acid peroxidation, and a same degree reduction (37%) in reduced glutathione levels (GSH), compared to the controls (Figure 4). Regarding the antioxidant enzymatic system, MAFLD resulted in a marked decline in the activities of catalase, GPx, and SOD by 46%, 26%, and 35%, respectively, compared to the controls (Table 5).

The administration of atorvastatin in MAFLD-induced rats achieved a moderate reduction in MDA levels by 24% and increased GSH by 35% compared to the MAFLD group (Figure 4). The statin also produced an inducing effect on the enzymatic activities of CAT, GPx, and SOD by 39%, 24%, and 17%, respectively, compared with the MAFLD group (Table 5). In the study of Tinkel et al., antioxidative effects of statins have been shown to be due to a complex modulation of redox enzymes’ activity and expression, including reduced expression of the NADPH oxidase and myeloperoxidase enzymes, induction of the antioxidant enzymes and upregulation of their activity, and reduction of biomarkers of oxidation. In addition, statins exhibit pleiotropic LDL-independent effects to decrease ROS formation [38]. According to clinical data, the activities of antioxidant-related enzymes (CAT, GSH-Px, SOD) significantly increased after 4 weeks of therapy in patients treated with atorvastatin [39]. In view of this, statins along with other medication classes with well-established protective effects such as ACE inhibitors and ARBs, remain the best pharmacotherapeutic approach currently available to influence oxidative stress-related atherosclerotic changes, in addition to their known clinical effects on reducing blood pressure and cholesterol.

There is, however, mounting evidence in the literature indicating the important role of plant-based alternatives and phytochemicals of different types and origins in combating oxidative stress [40,41], with an emphasis on polyphenolic compounds [42].

As demonstrated by a study, pretreatment with an ethanolic extract from flowers and leaves of six Iranian *Tanacetum* taxa, namely, *T. tabrisianum*, *T. sonboli*, *T. chiliophyllum*, *T. hololeucum*, *T. kotschyi*, and *T. budjnurdense*, at doses ranging from 10 to 100 µg/mL suppressed oxidative stress in hydrogen peroxide (H_2_O_2_)-treated K562 cells by increasing the intracellular glutathione (GSH), decreasing reactive oxygen species (ROS), glutathione peroxidase (GPx), and glutathione reductase (GR) activities [43].

According to another study, Mahmoodzadeh et al. found that pretreatment and posttreatment with 70% methanol extract of *T. parthenium* at doses of 80 and 120 mg/kg produced marked hepatoprotective effects on CCl4-induced liver injury in rats. The results of their study showed that LDL, total cholesterol, triglycerides, and glucose levels decreased compared to the control groups. In addition, the extract increased the levels of HDL and albumin and brought the activity of the transaminases AST and ALT, and the antioxidant enzymes SOD and GPx to near normal values. The authors concluded that *T. parthenium* extract has the ability to prevent enzyme leakage from cells and stabilize hepatocyte membranes. The hepatoprotective effects were probably due to the flavonoid-rich methanolic extract [44].

In the present study, ETB administered at both doses significantly restored the lowered antioxidant enzyme activity (Table 5) and GSH levels (Figure 4) and decreased MDA formation (Figure 4) in the rats with metabolic deteriorations. ETBhd had a more prominent effect on GSH levels compared to ETBld as indicated by the 43% increase, and on the CAT, GPx, and SOD enzymes increasing their activity by 70%, 29%, and 44%, compared to the untreated MAFLD rat group (Table 5). The marked protective effect of the extract in our tested model of oxidative liver injury is most likely attributable to its high content of polyphenolic compounds, as was found in the quantification UHPLC analysis.

Treatment with acarbose (15 mg/kg/day) is helpful to prevent the increase in oxidative stress and vascular dysfunction induced by hyperglycemia [45]. In a study by Wang et al. [46] a treatment with acarbose decreased mean amplitude of glycemic excursions (MAGE), body weight, and serum triglycerides and increased serum adiponectin without having a significant effect on oxidative stress.

In line with these findings, in our study, acarbose treatment from week 5 up until the end of the experiment significantly increased catalase activity by 48% in the animal group 6 (used as the positive control for diabetes type 2) as compared to the MAFLD group.

#### 2.4.4. Histopathological Examination

A histological examination of pancreas and liver sections from the experimental groups of rats is presented in Figure 5. Pancreas (A) and liver (B) tissues from the control group were with normal morphological architecture structures. MAFLD rats revealed pathological changes in the pancreas (C) with degenerative parenchymal alterations and liver (D) with centrilobular degenerative parenchymal amendments. Pancreas (E) and liver (F) tissues from MAFLD rats treated with acarbose presented a regenerated hepatocyte structure and pancreatic parenchyma. MAFLD rats treated with ETB revealed a regenerated hepatocyte structure (G) and pancreas (H) with hyperplastic islets of Langerhans and preserved cellular parenchyma.

## 3. Materials and Methods

### 3.1. Plant Material 

The leaves of *T. balsamita* were collected from a herbal garden (Belopoptsi village, Gorna Malina region) in Bulgaria at 700 m a.s.l. (42.67° N 23.77° E), during the full flowering stage in July 2022. The plant was identified according to Kuzmanov (2012) [47]. Voucher specimen was deposited at Herbarium Academiae Scientiarum Bulgariae (SOM 177 806).

### 3.2. Sample Extraction

The plant material (leaves) was dried in the laboratory for one week at room temperature (20–22 °C) and in 50% of relative humidity. Then, it was comminuted with a grinder (Rohnson, R-942, 220–240 V, 50/60 Hz, 200 W, Prague, Czech Republic) and the powder was stored in a dry and cool place until further analysis. Air-dried powdered leaves (50 g) were extracted with 80% MeOH (1:20 *w*/*v*) by sonication (80 kHz, ultra-sound bath Biobase UC-20C) for 15 min (×2) at room temperature. The extract was concentrated in vacuo, defatted with CH_2_Cl_2_, and lyophilized (lyophilizer Biobase BK-FD10P) to yield 5.95 g crude extract. The lyophilized extract of *Tanacetum balsamita* (ETB) was then used for pharmacological tests and UHPLC-DAD analyses.

Chemicals and reagents. Chlorogenic acid (2), orientin (3), rutin (4), hyperoside (5), luteolin 7-*O*-glucoside (6), kaempferol 3-*O*-rutinoside (7), 3,4-diCQA (8), 3,5-diCQA (9), isorhamnetin 3-*O*-glucoside (10), luteolin (15), and nepetin (16), were provided from Extrasynthese (Genay, France). Nneochlorogenic (1) and rosmarinic acids (11) were provided from Phytolab (Vestenbergsgreuth, Germany).

UHPLC-DAD analysis. The UHPLC-DAD analyses were carried out on a Thermo Scientific Dionex UltiMate 3000 analytical system (ThermoFisher Scientific, Inc., Waltham, MA, USA) equipped with a Dionex UltiMate 3000 RS Pump (LPG-3400RS), Dionex UltiMate 3000 RS Autosampler (WPS-3000TRS), Dionex UltiMate 3000 RS Column Compartment (TCC3000RS), and Dionex UltiMate 3000 Diode Array Detector (DAD-3000). The separation was achieved on an Acquity UPLC BEH C18 column (2.1 × 100 mm, 1.7 μm) (Waters), with mobile phase consisting of A water (with 0.1% formic acid) and B acetonitrile (0.1% formic acid). The used gradient was as follows: 0 min 5% B, 1 min 5% B, 20 min 30% B, 25 min 70% B, 33 min 95% B [14]. The system was then returned to the initial condition and equilibrated for over 6 min. The flow rate was 300 μL/min and the injection volume was 1 μL. The UV wavelengths were 310 nm (phenolic acids) and 360 nm (flavonoids). The entire system was controlled by Chromeleon software, version 7.2.

Quantitative analysis. The UHPLC analysis of phenolic acids (1, 2, 8, 9, 11, 12, 13, and 14) and flavonoids (3, 4, 5, 6, 7, 10, 15, and 16) was carried out using the external standard method. Standard calibrations of 1, 2, 5, 8, 11, 15, and 16 were established at five data points covering the concentration range of each analyte according to the level expected in the plant samples. Working solutions containing 0.2, 0.1, 0.05, 0.02, and 0.01 mg/mL of the assayed phenolic acids were prepared from a stock solution in methanol containing 0.5 mg/mL. Triplicate analyses were performed for each concentration, and the peak area was detected at 310 nm (for phenolic acids) and 360 nm (for flavonoids). Based on the similar UV spectra, the quantity of 3, 4, 7, and 10 was determined based on the calibration curve of hyperoside (5), 6 as luteolin (15), while 9, 12, 13, and 14 was quantified as 3,4-dicaffeoylquinic acid (9).

### 3.3. In Vitro Cytotoxicity Study

#### 3.3.1. Cell Lines and Culture Conditions

The antiproliferative activity of the *Tanacetum balsamita* extract was assessed in a human hepatocellular carcinoma cell line (HEP-G2) purchased from the German Collection of Microorganisms and Cell Cultures (DSMZ GmbH, Braunschweig, Germany). Cell cultures were cultivated in growth medium RPMI 1640 supplemented with 10% fetal bovine serum (FBS), 5% l-glutamine, and incubated under standard conditions of 37 °C and 5% humidified CO_2_ atmosphere.

#### 3.3.2. MTT Cell Viability Assay

Cell viability of the HEP-G2 cell culture was evaluated following a 72 h exposure to 5-fold serial dilutions of the ETB (concentration range 800–400 μg/mL) using a standard methodology for assessing cell viability known as the Mosmann MTT method. Exponential-phased cells were harvested and seeded (100 μL/well) in 96-well plates at the appropriate density (1.5 × 10^5^). After 24 h, cells were treated with different concentrations of the tested material and incubated for 72 h under standard conditions. Filter-sterilized MTT substrate solution (5 mg/mL in PBS) was added to each well of the culture plate. A further 2–4 h incubation allowed for the formation of purple insoluble formazan crystals, which were dissolved in isopropyl alcohol solution containing 5% formic acid. Absorbance was measured at 550 nm and collected absorbance values were blanked against MTT- and isopropanol solution and normalized to the mean value of untreated control (100% cell viability).

### 3.4. Animals and Study Design

#### 3.4.1. Animals

All procedures involving animals were approved by the Animal Care Ethics Committee from the Bulgarian Food Safety Agency and performed by users licensed by the Veterinary Faculty of the Forestry University. An ethical clearance (No. 346 of 28.02.2023) was issued.

The experiments were conducted according to the Guidelines for Animal Care. The rats were housed, maintained, and euthanized in accordance with the relevant international rules and recommendations as outlined in the European Convention for the Protection of Vertebrate Animals used for Experimental and other Scientific Purposes (ETS 123).

Fifty male Wistar rats at two months of age (150–180 g) were used. Rats were housed in Plexiglas cages (4 per cage) in a 12/12 light/dark cycle, under standard laboratory conditions (ambient temperature 20 ± 2 °C and humidity 72 ± 4%). All animals were purchased from the National Breeding Center, Sofia, Bulgaria, and allowed a minimum of 7 days to acclimate to the new conditions before the start of the study. Food and fresh drinking water were available ad libitum throughout the experimental period of 8 weeks.

#### 3.4.2. Acute Oral Toxicity of Extract from *T. balsamita* (ETB) in Male Wistar Rats

Acute oral toxicity was performed using the method of Lorke [48] on 4 male Wistar rats, 8 weeks old and weighing 150–160 g. The lyophilized ETB was dissolved in distilled water and administered once orally to 4 rats in a dose of 3000 mg/kg body weight and in a volume of 0.5 mL/100 g weight using a gastric tube. The animals were observed for symptoms of toxicity and death up to 24 h.

Two doses (low and high) based on the LD50 value of ETB were used for the repeated-dose experiment (8-weekly administration of ETB). The experiment was conducted on forty male Wistar rats randomly divided into 8 groups with 5 animals each.

#### 3.4.3. Experimental Design

This study aimed to investigate the effects of ETB in male normal rats and rats with induced metabolic dysfunction-associated fatty liver disease (MAFLD). This pathology was established by feeding rats with a high-calorie diet, described in detail in our previous experiments [49], and by administering once intraperitoneally a low dose (45 mg/kg/i.p.) of streptozotocin (STZ) [50].

The design of the experiment is depicted in Table 6. Forty male Wistar rats were randomly divided into eight groups of five animals (n = 5) as follows:

Healthy Wistar rats receiving normal food and water for 8 weeks:Group 1—control group animals, with free access to fresh water and normal pelleted food;Group 2—rats orally treated with a low dose of ETB (ETBld, or 150 mg/kg, which is 1/20 LD50) for 8 weeks;Group 3—rats given high dose of the extract (ETBhd, or 300 mg/kg, 1/10 LD50) for 8 weeks.

#### 3.4.4. Pathological Model, Wistar Rats with Induced MAFLD

Rats in this category were fed with a high-calorie diet (HC). On the 21st day, rats were challenged with 45 mg/kg, i.p. streptozotocin (STZ) dissolved in citrate buffer 0.1 M, pH 4.4, 15 min after the i.p. administration of nicotinamide (NA, 110 mg/kg bw) [51]. On the seventh day following STZ injection, rats with blood glucose concentration over 9 mmol/L were regarded as successfully established models and were randomized into the following groups with 5 animals each (n = 5):Group 4—MAFLD pathological control group;Group 5—MAFLD group treated orally once a day with atorvastatin (ATS, 5 mg/kg/day) [52] (from 5th to 8th week), as the positive control for dyslipidemia.Group 6—MAFLD rats treated orally once a day with acarbose (5 mg/kg) [53] (from 5th to 8th week) as the positive control for diabetes type 2 (DT2);Group 7—MAFLD rats treated with low-dose (150 mg/kg/p.o.) extract of *T. balsamita* (ETBld) for the whole period (from 1st to the end of the 8th week);Group 8—MAFLD rats treated with high-dose (300 mg/kg/p.o.) extract of *T. balsamita* (ETBhd) for the whole period;

Groups 4—8 additionally received 10% fructose (HF) solution for drinking.

The body weight of the animals was measured weekly with a laboratory scale. Blood glucose level, triglycerides, and cholesterol of the experimental animals were measured once a week for 8 weeks with test strips.

At the end of the experimental period after overnight starvation, the animals were sacrificed with a laboratory guillotine, blood was collected and biochemical parameters in serum were measured. Afterwards, the livers were taken to assess the oxidative stress biomarkers, malone dialdehyde (MDA) and reduced glutathione (GSH), and the activity of the antioxidant enzymes, glutathione peroxidase (GPx), catalase (CAT), and superoxide-dismutase (SOD). Small pieces from the livers and pancreas were taken and fixed in 10% buffered formalin for histopathological investigation.

#### 3.4.5. Assessment of Serum Biochemical and Hematological Parameters

The weekly measurement of blood glucose levels, triglycerides, and cholesterol was performed using a blood drop from the tail vein and multiparametric system for the monitoring of metabolic disorders MULTICAREIN (BSI Diagnostics, Arezzo, Italy).

At the end of the experimental period (8 weeks), whole blood was analyzed by a semi-automated hematological analyzer BC-2800 Vet, (Mindray, Shenzhen, China) according to the manufacturer’s instructions. The count of white blood cells (WBC), red blood cells (RBC), platelets (PLT), amount of hemoglobin (Hb), and hematocrit (Ht) were measured. The biochemical serum data as glucose (GLU), uric acid (UA), the activity of the enzymes amylase (AMYL), aspartate aminotransferase (ASAT), alanine aminotransferase (ALAT), and lipase, and the amount of total protein, albumin, total bilirubin, and direct bilirubin were measured using automated biochemistry analyzer kits (BS-120, Mindray, China), following the manufacturer’s instructions.

#### 3.4.6. Assessment of the Oxidative Stress Biomarkers

Oxidative damage was determined by measuring the quantity of thiobarbituric acid reactive substances (TBARS), expressed as malondialdehyde (MDA) equivalents as described by Polizio and Peña [54]. Reduced glutathione (GSH) was assessed by measuring the non-protein sulfhydryls after precipitation of proteins with trichloracetic acid (TCA) using the method described by Bump [55]. The antioxidant enzyme activities were measured in the supernatant of 10% homogenate, prepared in 0.05 M phosphate buffer (pH 7.4). The protein content of liver homogenate was measured by the method of Lowry [56]. Glutathione peroxidase (GPx) was measured by NADPH oxidation, using a coupled reaction system consisting of GSH, glutathione reductase (GR), and cumene hydroperoxide [57]. Catalase activity (CAT) was measured by the method described by Aebi [58]. Briefly, 10 µL of homogenate was added to 1990 µL of H_2_O_2_ solution. CAT activity was determined by measuring the decrease in absorbance at 240 nm. The enzyme activity was expressed as nmol/mg/min.

Superoxide dismutase activity (SOD) was measured according to the method of Misra and Fridovich [59], following spectrophotometrically the autoxidation of epinephrine at pH = 10.4 at 30 °C, using the molar extinction coefficient of 4.02 mM^−1^ cm^−1^.

#### 3.4.7. Histopathological Examination

Histopathological examination was performed using the method of Bancroft and Gamble [60]. The sections were observed under a high-power microscope and photomicrographs were taken using “Olympus” CX31 and Camera “Olympus x Optical zoom” with objective “PlanaC” 4/0.10 (Karl Zeiss, Oberkochen, Germany).

##### Statistical Analysis

Statistical analysis was performed by the MEDCALC program. Results are expressed as mean ± SD on five rats in each group. Experimental groups were compared by the Kruskal–Wallis variance analysis test, and a post hoc analysis using Bonferroni-corrected Mann–Whitney U test was performed. Values *p* ≤ 0.05 were considered statistically significant.

## 4. Conclusions

The global incidence of metabolic syndrome is rising at a rapid rate and is reaching alarming epidemic proportions, demanding the exploration of new treatment modalities for effective primary prevention of its co-associated morbidities. The present study provides the first in vivo evidence of the beneficial effects of a *Tanacetum balsamita* extract on lipid and glucose metabolism and antioxidant status in rat animal models with induced MAFLD. The high-dose extract produced a prominent reduction in triglyceride and cholesterol levels as well as lipase activity that was comparable to the reference drug atorvastatin. In terms of glucose homeostasis, the eight-week exposition to both the high- and low-dose phytotherapy additionally proved noninferior to the reference acarbose treatment, attributable to their common α-glucosidase and α-amylase inhibitory properties. The ETB was found to be rich in polyphenolic compounds such as 3,5-dicafeoylquinic, chlorogenic and rosmarinic acids, and provided a dose-dependent stimulatory effect on the in vivo antioxidant response through significant restoration of glutathione levels, catalase, glutathione peroxidase, and superoxide dismutase activity. Our findings are a testament to the complex and diverse health-promoting effects of the *Tanacetum balsamita* species which can be considered a viable approach in managing the related components of the metabolic syndrome and its complications as both an alternative and adjuvant means to conventional therapy.

## Figures and Tables

**Figure 1 plants-13-02086-f001:**
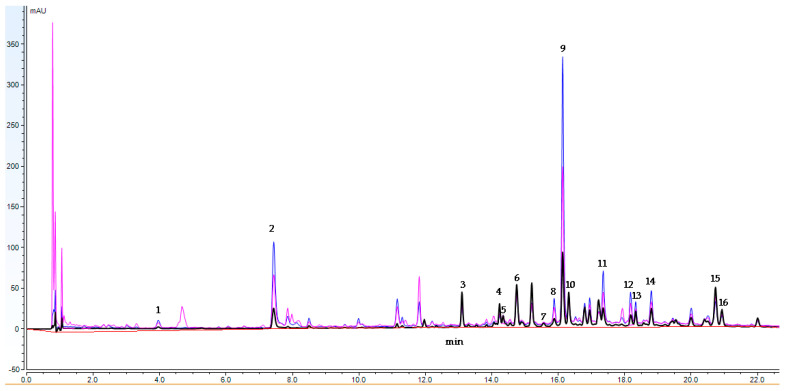
UHPLC-DAD chromatograms of extract from *Tanacetum balsamita* (ETB); wavelengths: 360 nm, 310 nm, and 280 nm.

**Figure 2 plants-13-02086-f002:**
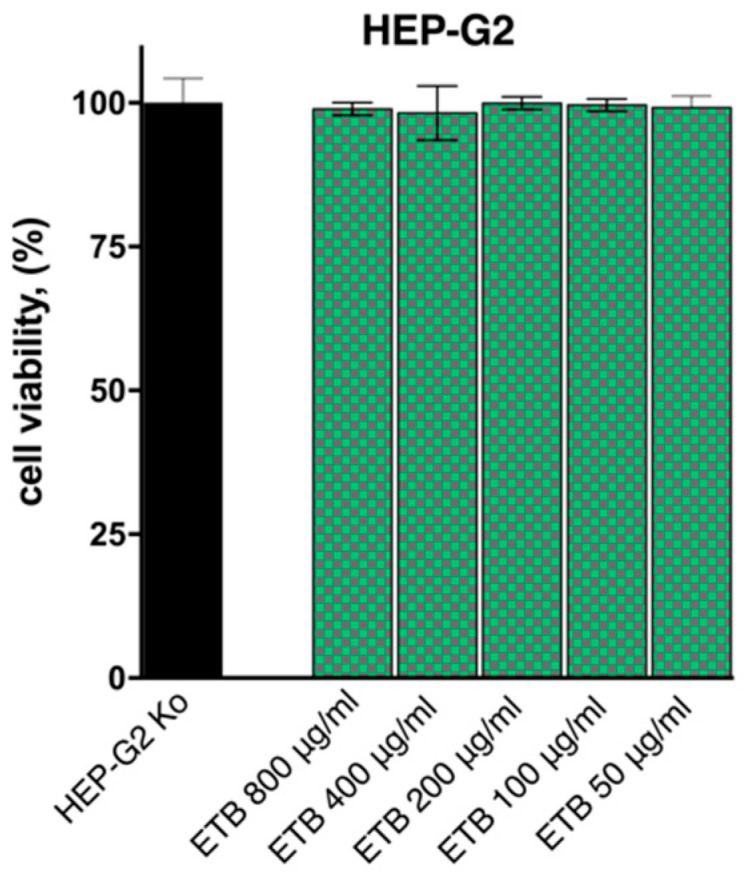
Cell viability of HEP-G2 cells following a 72 h exposure to various concentrations of the tested ETB extract compared to untreated control.

**Figure 3 plants-13-02086-f003:**
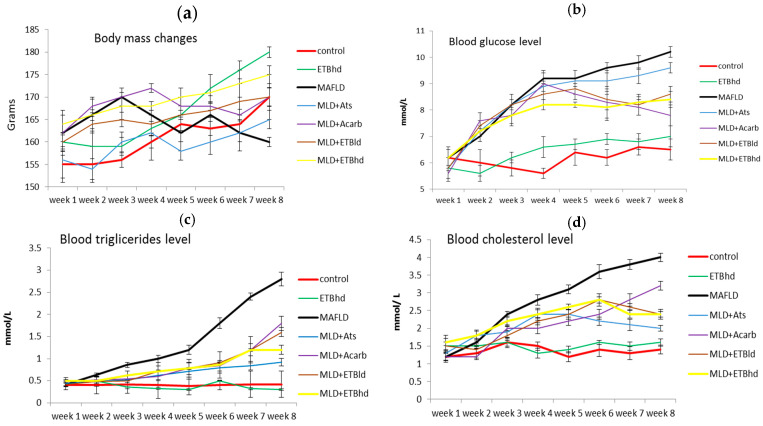
Weekly body weight changes (**a**), blood glucose level changes (**b**), blood triglicerides level (**c**), and blood cholesterol levels (**d**). Results are expressed as mean ± SD (n = 5). The significance of the data was assessed using the Bonferroni-corrected Mann–Whitney U test. Values of *p* ≤ 0.01 were considered statistically significant and presented in the Appendix A. Abbreviations: ETBld, extract of *T. balsamita*, low dose (150 mg/kg); ETBhd, extract of *T. balsamita* high dose (300 mg/kg); MAFLD, metabolic dysfunction-associated fatty liver disease; MLD+Ats, metabolic dysfunction-associated fatty liver disease+atorvastatin; MLD + Acarb, metabolic dysfunction-associated fatty liver disease+acarbose; MLD + ETBld, metabolic dysfunction-associated fatty liver disease + extract of *T. balsamita*, low dose (150 mg/kg); MLD + ETBhd, metabolic dysfunction-associated fatty liver disease + extract of *T. balsamita*, high dose (300 mg/kg).

**Figure 4 plants-13-02086-f004:**
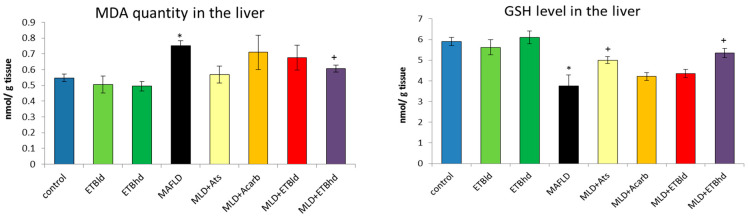
Levels of malonedialdehide (MDA) and reduced glutathione (GSH) in the liver homogenates. * *p* ≤ 0.01 vs. control; + *p* ≤ 0.01 vs. MAFLD. Results are expressed as mean ± SD (n = 5). The significance of the data was assessed using the Bonferroni-corrected Mann–Whitney U test. Values of *p* ≤ 0.01 were considered statistically significant.

**Figure 5 plants-13-02086-f005:**
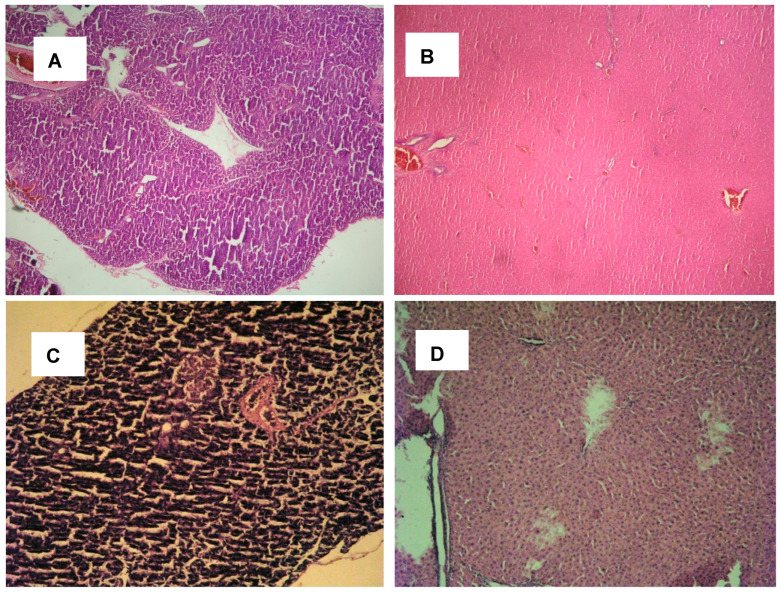
Histological profiles of pancreas and livers from the experimental groups. (**A**,**B**) Pancreas and liver tissues from control group; (**C**,**D**) MAFLD rats; (**E**,**F**) MAFLD rats treated with acarbose; (**G**,**H**) Pancreas and liver tissues from MAFLD rats treated with ETB (magnification 10 × 0.25).

**Table 1 plants-13-02086-t001:** Regression equation, limit of detection (LOD), and limit of quantification (LOQ) obtained for the main compounds in ETB.

№	Analyte	t_R_	UV	Equation	R^2^	LOD (μg/mL)	LOQ (μg/mL)
1	neochlorogenic acid	4.38	310	y = 0.6152x + 3.2153	0.9966	0.64	1.95
2	chlorogenic acid	7.70	310	y = 0.1025x + 4.977	0.9996	1.19	3.59
5	hyperoside	14.34	310	y = 0.1423x + 0.1789	0.9973	0.12	0.35
8	3,4-diCQA	15.91	310	y = 0.0939x + 0.1103	0.9996	0.19	0.57
11	rosmarinic acid	17.38	310	y = 0.103x + 0.3507	0.9995	0.96	2.91
15	luteolin	20.76	360	y = 0.2854x + 0.0108	0.9966	0.07	0.21
16	nepetin	20.94	360	y = 0.3126x + 0.6411	0.9890	0.07	0.22

**Table 2 plants-13-02086-t002:** Content (μg/mg dry extract) of compounds assayed in ETB.

No.	Analyte	t_R_	Content (μg/mg de)
1.	neochlorogenic acid	4.38	2.817 ± 0.297
2.	chlorogenic acid	7.70	17.967 ± 0.113
3.	orientin	13.10	4.603 ± 0.401
4.	hyperoside	14.23	3.922 ± 0.356
5.	rutin	14.34	0.483 ± 0.017
6.	luteolin 7-*O*-glucoside	14.75	3.671 ± 0.316
7.	kaempferol 3-*O*-rutinoside	15.59	0.857 ± 0.015
8.	3,4-diCQA	15.91	7.252 ± 0.921
9.	3,5-diCQA	16.17	69.129 ± 9.344
10.	isorhamnetin 3-*O*-glucoside	16.34	5.720 ± 0.584
11.	rosmarinic acid	17.38	12.950 ± 2.627
12.	AQA1	18.20	9.491 ± 0.250
13.	AQA2	18.35	6.586 ± 0.072
14.	AQA3	18.82	8.130 ± 0.505
15.	luteolin	20.76	4.049 ± 0.393
16.	nepetin	20.94	1.039 ± 0.141

**Table 4 plants-13-02086-t004:** Hematological parameters in all treated groups.

Hematology	Controls	ETBld	ETBhd	MAFLD	MLD + Ats	MLD + Acarb	MLD + ETBld	MLD + ETBhd	Ref. Range (Rats)
WBC × 10^3^/µL	8.75 ± 2.3 ^#^	8.65 ± 2.7 ^#^	8.63 ± 1.3 ^#^	12.25 ± 1.2 *^#^	10.63 ± 1.4	12.03 ± 2.6 ^#^	9.7 ± 1.4 ^+#^	8.12 ± 1.8 ^+^	1.96–8.25
RBC × 10^6^/µL	9.11 ± 0.17	9.00 ± 0.37	8.69 ± 0.69	9.05 ± 0.17	8.98 ± 0.22	8.58 ± 0.42	8.00 ± 0.64	8.55 ± 0.37	5.6–10.4
Hgb g/L	153.3 ± 8.8	157.2 ± 6.8	154.8 ± 6.1	153.5 ± 4.8	159.5 ± 8.1	151.3 ± 6.2	133.4 ± 4.5	144.5 ± 6.8	137–176
HCT %	43.15 ± 2.6	44.03 ± 1.9	44.83 ± 2.9	41.11 ± 1.2	45.33 ± 3.5	41.65 ± 1.1	40.15 ± 4.9	41.63 ± 1.6	39.6–52.5
PLT × 10^3^/ µL	437 ± 30.1 ^#^	487 ± 41.2 ^#^	395 ± 24.5 ^#^	618 ± 32.2 *^#^	426 ± 48.4 ^+#^	435 ± 24.5 ^+#^	608 ± 30.1 ^#^	430 ± 37.6 ^+#^	638–1177

* *p* ≤ 0.01 vs. control; ^+^
*p*≤ 0.01 vs. MAFLD; ^#^ *p* ≤ 0.01 vs. ref. Range. Results are expressed as mean ± SD (n = 5). The significance of the data was assessed using the Bonferroni-corrected Mann–Whitney U test. Values of *p* ≤ 0.01 were considered statistically significant. Abbreviations: WBC, white blood cells (leucocytes); RBC, red blood cells (erythrocytes); Hgb, hemoglobin; HCT, hematocrit; PLT, platelets.

**Table 5 plants-13-02086-t005:** Activity of antioxidant enzymes.

Antioxidant Enzymes	Controls	ETBld	ETBhd	MAFLD	MLD + Ats	MLD + Acarb	MLD + ETBld	MLD + ETBhd
Catalase nmol/mg/min	34.6 ± 1.6	44.1 ± 2.6 *	40.8 ± 1.7 *	18.7 ± 1.2 *	25.9 ± 2.9 ^+^	27.7 ± 1.2 ^+^	28.6 ± 0.98 ^+^	31.7 ± 1.2 ^+^
GPx nmol/mg/min	340.6 ± 11.9	344.9 ± 7.9	376.7 ± 15.3	251.3 ± 14.9 *	312.2 ± 11.3 ^+^	258.4 ± 26.7	300.3 ± 21.4 ^+^	323.2 ± 16.5 ^+^
SOD nmol/mg/min	280.7 ± 23.6	290.5 ± 10.6	325.2 ± 10.9 *	181.1 ± 14.3 *	211.6 ± 7.9	195.8 ± 6.1	251.8 ± 13.7 ^+^	259.9 ± 4.7 ^+^

* *p* ≤ 0.01 vs. control; ^+^
*p* ≤ 0.01 vs. MAFLD. Results are expressed as mean ± SD (n = 5). The significance of the data was assessed using the Bonferroni-corrected Mann–Whitney U test. Values of *p* ≤ 0.01 were considered statistically significant. Abbreviations: GPx, glutathione-peroxidase; SOD, superoxide-dismutase.

**Table 6 plants-13-02086-t006:** Experimental design.

Groups (n = 5)	Week 1	Week 2	Week 3	21st Day	Week 4	Week 5	Week 6	Week 7	Week 8
1.Control	Food and water ad libitum	Food and water ad libitum	Food and water ad libitum	Food and water ad libitum	Food and water ad libitum	Food and water ad libitum	Food and water ad libitum	Food and water ad libitum	Food and water ad libitum
2.ETBld	ETBld 150 mg/kg	ETBld 150 mg/kg	ETBld 150 mg/kg	ETBld 150 mg/kg	ETBld 150 mg/kg	ETBld 150 mg/kg	ETBld 150 mg/kg	ETBld 150 mg/kg	ETBld 150 mg/kg
3.ETBhd	ETBhd 300 mg/kg	ETBhd 300 mg/kg	ETBhdn 300 mg/kg	ETBhd 300 mg/kg	ETBhd 300 mg/kg	ETBhd 300 mg/kg	ETBhd 300 mg/kg	ETBhd 300 mg/kg	ETBhd 300 mg/kg
4. MAFLD	HC-HFD	HC-HFD	HC-HFD	NA-STZ (110/45 mg/kg bw, i.p.)	HC-HFD	HC-HFD	HC-HFD	HC-HFD	HC-HFD
5. MAFLD-atorvastat	HC-HFD	HC-HFD	HC-HFD	NA-STZ (110/45 mg/kg bw, i.p.)	HC-HFD	HC-HFD+ atorvastatin, 5 mg/kg/day	HC-HFD+ atorvastatin, 5 mg/kg/day	HC-HFD+ atorvastatin, 5 mg/kg/day	HC-HFD+ atorvastatin, 5 mg/kg/day
6. MAFLD-acarbose	HC-HFD	HC-HFD	HC-HFD	NA-STZ (110/45 mg/kg bw, i.p.)	HC-HFD	HC-HFD+ acarbose 5 mg/kg/po	HC-HFD+ acarbose 5 mg/kg/po	HC-HFD+ acarbose 5 mg/kg/po	HC-HFD+ acarbose 5 mg/kg/po
7. MAFLD-ETBld	HC-HFD + ETBld (150 mg/kg/po)	HC-HFD + ETBld (150 mg/kg/po/d)	HC-HFD + ETBld (150 mg/kg)	NA-STZ (110/45 mg/kg bw, i.p.)	HC-HFD + ETBld (150 mg/kg/po/)	HC-HFD + ETBld (150 mg/kg/po/d)	HC-HFD + ETBld (150 mg/kg/po/d)	HC-HFD + ETBld (150 mg/kg/po/d)	HC-HFD + ETBld (150 mg/kg/po)
8. MAFLD-ETBhd	HC-HFD + ETBhd (300 mg/kg/po/d)	HC-HFD + ETBhd (300 mg/kg/po/d)	HC-HFD + ETBhd (300 mg/kg/po/d)	NA-STZ (110/45 mg/kg bw, i.p.)	HC-HFD	HC-HFD + ETBhd (300 mg/kg/po/d)	HC-HFD + ETBhd (300 mg/kg/po/d)	HC-HFD + ETBhd (300 mg/kg/po/d)	HC-HFD + ETBhd (300 mg/kg/po/d)

Abbreviations: ETBld, low dose of extract from *T. balsamita*; ETBhd, high dose of extract from *T. balsamita*; MAFLD, metabolic dysfunction-associated fatty liver disease; HC-HFD, high calorie-high fructose diet; NA, nicotinamide; STZ, streptozotocin.

## Data Availability

The original contributions presented in the study are included in the article.

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
