# Peer review of "Mitigating Effects of Tanacetum balsamita L. on Metabolic Dysfunction-Associated Fatty Liver Disease (MAFLD)"

_plants, 2024, doi:10.3390/plants13152086_

Round 1
Reviewer 1 Report
Comments and Suggestions for Authors
1. One of the pictures in Figure 1 could be deleted.
2. There were 16 compounds in table 2, but only 7 compounds had Regression equation. How were the other 9 compounds quantified?
3. The expression is not concise enough or the format is incorrect, for example, in Line 218 the explanation about lipase; Line 252; the title of Figure 4.
4. The words "liver oxidative stress markers" in the title of Table 5 should be better deleted.
5. The frequency of most ultra-sound baths is 40kHz. The author should check the frequency of the bath with 80kHz in the manuscript.
Comments on the Quality of English Language
The text can be more concise.
Author Response
- One of the pictures in Figure 1 could be deleted.
Response: Dear reviewer, thank you for your remark. The correction has been made.
- There were 16 compounds in table 2, but only 7 compounds had Regression equation. How were the other 9 compounds quantified?
Response: Thank you for your question. Based on the similar UV spectra, the quantity of 3, 4, 7, and 10 was determined based to the calibration curve of hyperoside (5), 6 as luteolin (15), while 9, 12, 13, 14 was quantified as 3,4-dicaffeoylquinic acid (9).
- The expression is not concise enough or the format is incorrect, for example, in Line 218 the explanation about lipase; Line 252; the title of Figure 4.
Response: Thank you for your note. The caption of Figure 4 has been improved.
- The words "liver oxidative stress markers" in the title of Table 5 should be better deleted.
Response: Thank you for your note. The caption of Table 5 has been improved.
- The frequency of most ultra-sound baths is 40kHz. The author should check the frequency of the bath with 80kHz in the manuscript.
Response: Thank you for the remark. It was checked. The model of ultra‐sound bath that we used in the study is Biobase UC‐20C with frequency of 80kHz .
Reviewer 2 Report
Comments and Suggestions for Authors
In the paper “Mitigating effect of Tanacetum balsamita L. on metabolic dysfunction-associated fatty liver disease (MAFLD)”, Mihaylova and collaborators, describes the beneficial effect of the Tanacetum balsamita methanol-aqueous extract on biochemical parameters related to oxidative stress, metabolic syndrome and liver function in an animal model of MAFLD. In my opinion the study reaches a good level of novelty, despite I suggest Authors to review the editing of English phrasing.
In my opinion, the paper can be considered for publication only after major revisions listed below:
1) Between lines 34-41, the word “associated” is repeated several times, thus I suggest Authors to replace this word with others or reorganize the sentence.
2) The writing of this paper is characterized by very long periods, making poorly fluent the text and limiting what the authors want to communicate. For this reason, I suggest Authors to reorganize the senteces of this paper, particularly in the introduction paragraph.
3) In line 78 Authors should add a reference about the antioxidant properties of Tanacetum species.
4) In the paragraph 2.3.1, it has been reported that animals of MAFLD group, showed a reduction of body weight. Are the Authors able to deeply demonstrate why this happens? I suggest to add more references in order to corroborate the teory by which patients affected by MAFLD could be not obese.
5) In the paragraph 2.3.1. Authors reported that animals of MAFLD group, showed a reduction of body weight, particularly in the last weeks of the experiment. Could the Authors explain why the body weight of animals belonging to MAFLD group is lower than animals of the control group?
6) Finally, I suggest Authors to improve the graphic style of the tables.
Comments on the Quality of English LanguageI suggest Authors to review the editing of English phrasing
Author Response
In the paper “Mitigating effect of Tanacetum balsamita L. on metabolic dysfunction-associated fatty liver disease (MAFLD)”, Mihaylova and collaborators, describes the beneficial effect of the Tanacetum balsamita methanol-aqueous extract on biochemical parameters related to oxidative stress, metabolic syndrome and liver function in an animal model of MAFLD. In my opinion the study reaches a good level of novelty, despite I suggest Authors to review the editing of English phrasing.
Response: Thank you for your comment. The manuscript has been edited by a knowledgeable language practitioner.
In my opinion, the paper can be considered for publication only after major revisions listed below:
1) Between lines 34-41, the word “associated” is repeated several times, thus I suggest Authors to replace this word with others or reorganize the sentence.
Response: Thank you for your suggestion. The correction has been made.
2) The writing of this paper is characterized by very long periods, making poorly fluent the text and limiting what the authors want to communicate. For this reason, I suggest Authors to reorganize the senteces of this paper, particularly in the introduction paragraph.
Response: Thank you for the suggestion. The text has been edited in terms of structure and clarity.
3) In line 78 Authors should add a reference about the antioxidant properties of Tanacetum species.
Response: Thank you very much for the comment. Several relevant citations were added in the text.
4) In the paragraph 2.3.1, it has been reported that animals of MAFLD group, showed a reduction of body weight. Are the Authors able to deeply demonstrate why this happens? I suggest to add more references in order to corroborate the teory by which patients affected by MAFLD could be not obese.
Response: Thank you for your suggestion. NAFLD is usually associated with type 2 DM, dyslipidemia, metabolic syndrome, and obesity. The risk factor for NAFLD is often considered to be obesity, but it can also occur in people with lean type, which is defined as lean NAFLD. [Chen M, Cao Y, Ji G, Zhang L. Lean nonalcoholic fatty liver disease and sarcopenia. Front Endocrinol (Lausanne). 2023 Jun 23;14:1217249. doi: 10.3389/fendo.2023.1217249.] This disease has mostly been studied in obese individuals; however, it has been widely reported and studied among the lean/non-obese population in recent years. The pathogenesis of NAFLD in non-obese patients is associated with various genetic predispositions, particularly a patatin-like phospholipase domain-containing protein 3 G allele polymorphism, which results in the accumulation of triglyceride in the liver and resistance to insulin. Additionally, dietary factors such as high fructose consumption seem to play a substantial role in the pathology of non-obese NAFLD. [Ahadi M, Molooghi K, Masoudifar N, Namdar AB, Vossoughinia H, Farzanehfar M. A review of non-alcoholic fatty liver disease in non-obese and lean individuals. J Gastroenterol Hepatol. 2021 Jun;36(6):1497-1507. doi: 10.1111/jgh.15353;
The prevalence of lean/non-obese NAFLD is around 20% within the NAFLD population, and 5% within the general population. Recent data suggest that individuals with lean NAFLD, despite the absence of obesity, exhibit similar cardiovascular- and cancer-related mortality compared to obese NAFLD individuals and increased all-cause mortality risk. [Kuchay MS, Martínez-Montoro JI, Choudhary NS, Fernández-García JC, Ramos-Molina B. Non-Alcoholic Fatty Liver Disease in Lean and Non-Obese Individuals: Current and Future Challenges. Biomedicines. 2021 Sep 28;9(10):1346. doi: 10.3390/biomedicines9101346;
Zhang Y, Xiang L, Qi F, Cao Y, Zhang W, Lv T and Zhou X (2024) The metabolic profiles and body composition of non-obese metabolic associated fatty liver disease. Front. Endocrinol. 15:1322563. doi: 10.3389/fendo.2024.1322563
Hu PF, Zeng X, Zou ZY, Tang W, Guo YB, Yuan ZL, Shi PM, Tan Y, Song Y, Shi YQ, Xie WF. The presence of NAFLD in nonobese subjects increased the risk of metabolic abnormalities than obese subjects without NAFLD: a population-based cross-sectional study. Hepatobiliary Surg Nutr. 2021 Dec;10(6):811-824. doi: 10.21037/hbsn-20-263]
Albhaisi S, Chowdhury A, Sanyal AJ. Non-alcoholic fatty liver disease in lean individuals. JHEP Rep. 2019 Aug 30;1(4):329-341. doi: 10.1016/j.jhepr.2019.08.002]
Chan WK. Comparison between obese and non-obese nonalcoholic fatty liver disease. Clin Mol Hepatol. 2023 Feb;29(Suppl):S58-S67. doi: 10.3350/cmh.2022.0350. Epub 2022 Dec 5. PMID: 36472052; PMCID: PMC10029940.
Phipps, Meaghan, and Julia Wattacheril. "Non-alcoholic fatty liver disease (NAFLD) in non-obese individuals." Frontline Gastroenterology 11.6 (2020): 478-483.]
5) In the paragraph 2.3.1. Authors reported that animals of MAFLD group, showed a reduction of body weight, particularly in the last weeks of the experiment. Could the Authors explain why the body weight of animals belonging to MAFLD group is lower than animals of the control group?
Response: Thank you for your question. The high-calorie food with which the animals were fed in this experiment was prepared at the Department of Drug Technology in the Faculty of Pharmacy of Medical University of Sofia and was described in detail in our previous publication (Petrova, A.; Simeonova, R.; Voycheva, C.; Savov, Y.; Marinov, L.; Balabanova, V.; Gevrenova, R.; Zheleva-Dimitrova, D. Meta-bolic syndrome: Comparison of three diet-induced experimental models. Pharmacia 2023, 70, 1539-1548. https://doi.org/10.3897/pharmacia.70.e109965.). This food differs in taste and consistency from the normal pelleted food that rats usually eat resulting in reduced food intake. However, this, together with the additional intake of fructose and streptozotocin, is enough to develop liver damage of the non-alcoholic steatohepatitis type.
6) Finally, I suggest Authors to improve the graphic style of the tables.
Response: As recommended, it has been improved.
7) Comments on the Quality of English Language. I suggest Authors to review the editing of English phrasing
Response: Thank you for your suggestion. The manuscript has been edited by a knowledgeable language practitioner.
Reviewer 3 Report
Comments and Suggestions for Authors
The study is very well-conceived, research-based and scientifically current, and primarily addresses the issue of health conditions related to non-communicable diseases of modern humanity, which above all has a potential impact on the wider, not only scientific, public.
All parts of the work are clearly and concisely conceived. In the Introduction, the problem that the research deals with is clearly presented, and other researches that deal with a similar topic are also concisely and in detail listed. The results are clearly presented, and all tables and graphs are necessary for their clear presentation. The discussion is adequate in most parts of the work, except for some segments for which I suggest the addition that I have listed in the text below. The material and methods are clearly described and provide all the necessary data so that the research can be repeated. The conclusions are clearly defined, but there are several suggestions for their improvement, which I have also listed in the text below.
Finally, I suggest this paper for publication with a few minor suggestions and comments for corrections, as follows:
Too many key words, I suggest listing only those not mentioned in the manuscript title
Results and discussion
Lines 94-98 superfluous as it is a description of the method and not the results
2.1. UHPLC-DAD analysis – the authors describe the results, but these results are not discussed, i.e. not compared with similar studies, etc.
pay attention to the citation of the literature, in some parts of the text it is not cited according to the journal requirements, e.g. line 177, 252, 353...
detailed descriptions of some tables and graphs, e.g. Figure 3, Table 3, Table 4, Figure 4, Table 5 - authors are advised to shorten the descriptions as much as possible, as this type of text is confusing and it is not possible to distinguish which part belongs to the description of tables or graphs and which part belongs to the description of the results and discussion.
I suspect it is a technical error, considering how Figure 5 has been inserted into the part of the text that refers to the description of the plant material (line 424)
Sample collection – line 431 - should be described in more detail how the leaves used in the experiment were dried, i.e. specify the conditions, approximate temperature during air drying, relative humidity, area where drying was performed. The preparation of the powder consistency should also be described, what equipment was used for grinding and sewing.
Line 433- please use the index in the chemical formula
Conclusions
Line 607 – please be more specific about the dose of extract used, give the exact concentration of extract dosed. What does a “significant reduction” mean? It is better to give the exact reduction, for example as a percentage.
Line 609-610 – “Exposure to high and low dose phytotherapy” – it is necessary to specify the exact dose used
Author Response
The study is very well-conceived, research-based and scientifically current, and primarily addresses the issue of health conditions related to non-communicable diseases of modern humanity, which above all has a potential impact on the wider, not only scientific, public.
All parts of the work are clearly and concisely conceived. In the Introduction, the problem that the research deals with is clearly presented, and other researches that deal with a similar topic are also concisely and in detail listed. The results are clearly presented, and all tables and graphs are necessary for their clear presentation. The discussion is adequate in most parts of the work, except for some segments for which I suggest the addition that I have listed in the text below. The material and methods are clearly described and provide all the necessary data so that the research can be repeated. The conclusions are clearly defined, but there are several suggestions for their improvement, which I have also listed in the text below.
Finally, I suggest this paper for publication with a few minor suggestions and comments for corrections, as follows:
- Too many key words, I suggest listing only those not mentioned in the manuscript title
Response: Thank you for the suggestion. Your recommendation has been taken into account.
Results and discussion
- Lines 94-98 superfluous as it is a description of the method and not the results
Response: Thanks for the comment. The text has been corrected.
- 1. UHPLC-DAD analysis – the authors describe the results, but these results are not discussed, i.e. not compared with similar studies, etc.
Response: Thank you for the comment. Discussion is embedded into the text: “The presence of the above mentioned compounds in costmary was reported previously, but without thorough quantitative data [12, 13]. However, a former HPLC-DAD analysis identified 4 major flavonoids in T. balsamita extract as quercetin, apigenin 7-O-glucoside (cosmosiin), luteolin 7-O-glucoside and luteolin 3-methyl ether (chrysoeriol) [16]. Additionally, cichoric acid was determined in that study as prevailing compound in the costmary, being present at 3.33 g/100 g extract. In contrast, cichoric acid was not found in our study. The values of chlorogenic and rosmarinic acid determined by Bazek and co-workers were considerably lower than those found in our study, while a lesser amount of luteolin 7-O glucoside was quantified.”
[16] Baczek KB, Kosakowska O, Przybyl JL, Pióro-Jabrucka E, Costa R, Mondel¬lo L, Gniewosz M, Synowiec A, Węglarz Z (2017) Antibacterial and anti¬oxidant activity of essential oils and extracts from costmary (Tanacetum balsamita L.) and tansy Tanacetum vulgare L.). Industrial Crops and Products 102: 154–163. https://doi.org/10.1016/j.indcrop.2017.03.009
4) pay attention to the citation of the literature, in some parts of the text it is not cited according to the journal requirements, e.g. line 177, 252, 353...
Response: Thank you for your note. Citation style has been updated.
- detailed descriptions of some tables and graphs, e.g. Figure 3, Table 3, Table 4, Figure 4, Table 5 - authors are advised to shorten the descriptions as much as possible, as this type of text is confusing and it is not possible to distinguish which part belongs to the description of tables or graphs and which part belongs to the description of the results and discussion.
Response: Thank you for your note. The caption text for figures and tables has been improved.
- I suspect it is a technical error, considering how Figure 5 has been inserted into the part of the text that refers to the description of the plant material (line 424)
Response: Thank you for your note. Errors in page layout have been fixed.
- Sample collection – line 431 - should be described in more detail how the leaves used in the experiment were dried, i.e. specify the conditions, approximate temperature during air drying, relative humidity, area where drying was performed. The preparation of the powder consistency should also be described, what equipment was used for grinding and sewing.
Response: Thank you very much for the recommendation. The description of the plant material (conditions, approximate temperature during air drying, relative humidity and etc) is added in the text: “The plant material (leaves) was dried in the laboratory for one week at room temperature (20-22oC) and 50% of relative humidity. Then, it was comminuted with a grinder (Rohnson, R-942, 220–240 V, 50/60 Hz, 200 W, Prague, Czech Republic) and the powder was stored in a dry and cool place until further analysis.”
- Line 433- please use the index in the chemical formula
Response: The correction has been made.
Conclusions
- Line 607 – please be more specific about the dose of extract used, give the exact concentration of extract dosed. What does a “significant reduction” mean? It is better to give the exact reduction, for example as a percentage.
Response: Thank you for your note. The two dosing regimens (high and low dose treatment concentrations) are specified in the text.
Line 609-610 – “Exposure to high and low dose phytotherapy” – it is necessary to specify the exact dose used
Response: Thank you for your note. The two dosing regimens (high and low dose treatment concentrations) are specified in the text.
Reviewer 4 Report
Comments and Suggestions for Authors
Dear authors,
I found your work really interesting and your results very impecive. I would like to suggest you some changes in order to make your work better.
Fisrt of all you should check the template of the journal. I think that the captions of the figures and tables are not in accordance to the template.
You should write in vivo and in vitro in italics mode everytime.
As it concerns the mai part and the topic I thnik that is related to the scope of the special issue but I think that it fits better the thematology of other journals because the part of your work rlated to the biochemical and pharmacological data is uncomperable to the phytochemical once.
In the part material and method you should check the instructions of the journal.
I think that Figure 5 will fit better before material and methods.
Moreover, I think that you have report that the quantitative and qualitative analysis of the extract will differ because the phytochemical contet is related to the extraction method but Tanacetum spp are well known for their high contet of phenolics according bibliography.
Author Response
Dear authors,
I found your work really interesting and your results very impecive. I would like to suggest you some changes in order to make your work better.
- Fisrt of all you should check the template of the journal. I think that the captions of the figures and tables are not in accordance to the template.
Response: Thank you for your note. The caption of figures and tables have been modified according to the journal template.
- You should write in vivo and in vitro in italics mode everytime.
Response: Thank you for the remark. The phrases have been converted from regular to italic text.
- As it concerns the mai part and the topic I thnik that is related to the scope of the special issue but I think that it fits better the thematology of other journals because the part of your work rlated to the biochemical and pharmacological data is uncomperable to the phytochemical once.
Response: With recent exponential development of metabolite and biological profiling, in-depth studies on T. vulgare, T. macrophyllum, T. balsamita, T. parthenium and T. poteriifolium bringh a more extended view on the secondary metabolites and the mode of action of the taxa. In addition to evoking an antioxidant response, Tanacetum extracts/isolated compounds displayed inhibitory activity towards enzymes involved in carbohydrate metabolism and neurotransmission, which generates further interest in the species. It is worth noting that further in vivo studies are needed to evaluate health-promoting application of Tanacetum extracts and isolated metabolites in pharmaceutical scale. Thus, this study is a part of our ongoing investigation of Tanacetum balsamita and its prominent activity in metabolic-associated fatty liver disease.
- In the part material and method you should check the instructions of the journal.
Response: Thank you for the recommendation. The journal guidelines have been appropriately followed.
- I think that Figure 5 will fit better before material and methods.
Response: Page and illustrations layout issues haven been resolved.
- Moreover, I think that you have report that the quantitative and qualitative analysis of the extract will differ because the phytochemical contet is related to the extraction method but Tanacetum spp are well known for their high contet of phenolics according bibliography.
Response: Thank you for the comment. In both our researches, Gervrenova et al. (2023) and the present study, the same plant material was analyzed subjected to the same extraction procedure. Therefore, the previously reported quantitative and qualitative data are applicable and valid.
Round 2
Reviewer 2 Report
Comments and Suggestions for Authors
The manuscript, can be considered for pubblication
Author Response
Thank you for the comment.
Reviewer 4 Report
Comments and Suggestions for Authors
Accept in present form
Author Response
Thank you for the comment.